# Link Prediction on Multilayer Networks through Learning of Within-Layer and Across-Layer Node-Pair Structural Features and Node Embedding Similarity

## ABSTRACT

Link prediction has traditionally been studied in the context of simple graphs, although real-world networks are inherently complex as they are often comprised of multiple interconnected components, or layers. Predicting links in such network systems, or *multilayer networks*, require to consider both the internal structure of a target layer as well as the structure of the other layers in a network, in addition to layer-specific node-attributes when available. This problem poses several challenges, even for graph neural network based approaches despite their successful and wide application to a variety of graph learning problems. In this work, we aim to fill a lack of multilayer graph representation learning methods designed for link prediction. Our proposal is a novel neural-network-based learning framework for link prediction on (attributed) multilayer networks, whose key idea is to combine (i) pairwise similarities of multilayer node embeddings learned by a graph neural network model, and (ii) structural features learned from both within-layer and across-layer link information based on overlapping multilayer neighborhoods. Extensive experimental results have shown that our framework consistently outperforms both single-layer and multilayer methods for link prediction on popular real-world multilayer networks, with an average percentage increase in AUC up to 38%. **We make source code and evaluation data available to the research community at https://shorturl.at/cOUZ4.**

## CCS CONCEPTS

• **Computing methodologies → Machine learning**.

## KEYWORDS

graph representation learning, graph neural networks, link prediction, multilayer networks

**ACM Reference Format:**
Anonymous Author(s). 2018. Link Prediction on Multilayer Networks through Learning of Within-Layer and Across-Layer Node-Pair Structural Features and Node Embedding Similarity. In *Proceedings of Make sure to enter the correct conference title from your rights confirmation emai (Conference acronym 'XX)*. ACM, New York, NY, USA, 15 pages. https://doi.org/XXXXXXX.XXXXXXX

## 1 INTRODUCTION

A wide range of complex network systems, from online social networks (OSNs) to transportation networks and biological networks, can profitably be modeled using the formalism of *multilayer networks* [22], in which two or more layers are interconnected and represent different types or contexts of relationships between the entities. For instance, in the landscape of OSNs, layers can represent different types of interaction (e.g., mention, like/favorite) for the same user base, but also different platforms providing multiple connectivity contexts for the same users. Modeling such complex network systems with a flattened graph or multiple independent graphs would lead to a loss of important information, since both internal, i.e., *within-layer*, and external, i.e., *across-layer*, structural features are needed to discover knowledge patterns from a complex network [6, 25, 32, 33].

Like in simple graphs, one fundamental problem in multilayer networks is *link prediction*, which is to estimate the likelihood that a link exists between two nodes in one of the layers, based on structural information involving the nodes within and outside that layer; again, in the OSN domain, an intuitive application of link prediction is suggesting new friends for users on one target platform (i.e., layer), by also exploiting the social contacts users have on other platforms. Surprisingly, despite the pervasiveness of multilayer networks in real-world scenarios, most existing studies on link prediction have focused on single-layer networks.

Over the last few years, neural networks (NNs) and especially graph neural networks (GNNs) have become the dominant paradigm for machine learning on single-layer networks. While GNN-based frameworks have achieved strong performance for tasks such as node and graph classification, their representation learning design hinders GNNs in capturing link-specific information; as a consequence, classic heuristics might achieve comparable performance with GNNs for the link prediction task. Indeed, GNNs are not effective in distinguishing automorphic nodes (i.e., nodes having the same structural role in the graph) and cannot focus on information relevant for target pair(s) of nodes (e.g., overlapping neighborhoods) [47]. The latter limitation is particularly crucial for link prediction, as it requires to properly model pairwise (common) neighbors of nodes, whereas GNNs heavily rely on smoothed node-features rather than graph structure [42]. To alleviate this issue for link prediction in simple graphs, some methods aim to inject link structural information in the learning process [10, 41, 42, 45].

In this work, we propose a neural-network-based learning framework for link prediction in (attributed) multilayer networks, which brings the aforementioned idea of incorporating structural information at node-pair level into the learning process for link prediction. To compute the likelihood of existence of a link between two nodes

on a specific layer, our proposed approach jointly learns two components, namely *GNN-based node embedding* and *NN-based node-pair neighborhood feature extraction.* The former relies on node-level graph representation learning methods specifically designed for multilayer networks, and as such is able to integrate available external information associated with nodes (i.e., node attributes or initial features); the latter is designed to extract node-pair-level structural features based on shared neighborhoods of any two nodes in different layers. Both components leverage within-layer as well as across-layer information to contribute to computing the likelihood of existence of a link, but in different ways: the GNN-based node embedding component considers information across all layers according to the message passing paradigm in multilayer networks, whereas the NN-based node-pair neighborhood feature extraction component accounts for different notions of *overlapping multilayer neighborhood* between nodes located in different layers.

We summarize our contributions as follows:

(1) We define a neural-network-based learning framework for link prediction on (attributed) multilayer networks, named ML-Link. To the best of our knowledge, we are the first to propose augmenting multilayer GNNs with node-pair features learned from both within-layer and across-layer structural information.

(2) ML-Link leverages different types of overlapping multilayer neighborhoods and adaptively incorporates their information using an attention mechanism to generate node-pair structural features for link prediction.

(3) Experimental evaluation on real-world and synthetic multilayer networks has shown the significance of ML-Link and its superiority of against 11 competing methods and 6 baselines, with percentage increase in AUC averaged over all competitors ranging from 5% to 38%. Also, results indicate the usefulness of all architectural components of ML-Link, and a certain robustness w.r.t. the main hyper-parameter.

## 2 RELATED WORK

**Link prediction on simple networks.** Link prediction has been traditionally addressed using heuristic methods or latent-feature methods [44]. The former generally determine the likelihood of links based on topological measures of node similarity, such as Common neighbors [26], Jaccard score, Preferential attachment [3], Adamic-Adar [1], Resource Allocation [48] and SimRank [20]. Latent-feature methods extract node vectorial representations from the graph structure through matrix factorization, and apply the inner product to pairs of such representations to predict links [44]. Also, shallow graph-representation learning methods, e.g., LINE [38], DeepWalk [35], and node2vec [17], can be used to learn node representations based on random walks.

GNN methods for link prediction can be categorized into subgraph-based and node-based approaches [44]. Subgraph-based methods [8, 39, 46], exemplified by SEAL [45], extract local subgraphs around each target link and use GNNs to map subgraph patterns to link existence. While usually outperforming node-based methods, subgraph-based methods may suffer from high computational complexity [44]. Node-based methods, pioneered by Graph AutoEncoder (GAE) [21], learn node embeddings from local neighborhoods and aggregate

them using GNNs to construct link representations. Node-based methods can perform worse than traditional heuristics due to their dependence on smoothed node features rather than graph structure. Neo-GNN [42] addresses this limitation by learning structural features from an adjacency matrix and estimating overlapping neighborhoods for link prediction.

Our proposed ML-Link adopts the core idea of Neo-GNN, which is to exploit structural features based on overlapping neighborhoods. However, we extend this to the multilayer link prediction setting, based on multilayer neighborhood definitions, and by relying on GNNs specifically conceived for multilayer networks. Moreover, ML-Link is highly versatile and modular in terms of the adopted GNN model since, should better GNNs for multilayer networks be devised in the future, these can easily be incorporated into our approach by just changing one building block.

**Link prediction on multilayer networks.** Link prediction methods for multilayer networks consider information from some or all layers to predict the likelihood of a link between two nodes in a specific layer. Early works include [36], which uses a collection of heuristic link-prediction scores computed for each relation type as input features for a decision tree (in addition to multiplex features such as the score average and entropy across layers); [18], which aggregates different traditional scores to feed a random forest classifier; [19], which applies a support vector machine (SVM) with a Gaussian kernel to features extracted from a metagraph built upon the application of a community detection algorithm on each link type separately. Such methods were mainly evaluated on specific types of multilayer network, namely bibliography data ([36]), geo-social data ([18]), and two-layer social networks ([19]).

A generalization of the Adamic-Adar method for multiplex networks is given in [2], although without considering that two nodes could be connected in other layers than the target one. MELL [30] embeds each layer into a low dimensional space, capturing the shared connectivity across layers. CrossMNA [12] leverages across-layer information by jointly learning an intra-vector and an inter-vector for each node; the former can be used for link prediction, and the latter for node matching across the layers. MAGMA [14] derives graph association rules by identifying all frequent patterns in a network via multiplex graph mining, then it assigns a score to each disconnected node-pair by finding the occurrences of each rule in the network. [43] proposes one of the earliest GNN-based framework designed for attributed multilayer networks, as it learns node representations by exploiting both intra- and inter-layer dependencies. GATNE [9] is designed for attributed multiplex heterogeneous networks, which also include the particular case of a multiplex network with a single type, hence GATNE can be used for link prediction on homogeneous multiplex networks.

**Comparison with our ML-Link.** Unlike the GNN-based multilayer methods proposed in [43] and [9], we enhance the GNN predictive ability by computing structural features extracted from multilayer interactions, which are beneficial for the link prediction task where the topological aspect is crucial [29]. Indeed, the ability of GNNs to incorporate topological features is insufficient for the task of link prediction, and simple heuristics like Adamic-Adar or Jaccard indexes can sometimes achieve better performance by a large margin [42, 45, 47].

Compared to [12] and [30], our approach can leverage (external) node-attributes at each layer. Additionally, those methods exploit across-layer interactions by preserving the similarity of the embeddings associated with the same nodes across different layers, which could be suboptimal in cases where the structure of the layers is substantially different. In contrast, our method for harnessing multi-layer dependencies is custom-designed to address the link prediction task.

Differently from [18, 36], we do not aggregate multiple scores derived from traditional heuristics for link prediction (e.g., Adamic-Adar) nor we utilize predefined heuristics for structural feature extraction [2, 19]; rather, we learn structural features of nodes in a data-driven fashion from each layer, and use them to generate different link existence scores for each pair of nodes, encompassing both within-layer and across-layer interactions. We also jointly leverage different notions of multilayer neighborhoods in order to get a more holistic view of the interplay between layers.

Compared to [14] whereby associative rules are extracted considering the entire multiplex network, our approach explicitly differentiates between the contribution provided by within-layer and across-layer interactions. Yet, generating associative rules can potentially be burdensome due to the extraction of frequent patterns across the entire network.

To the best of our knowledge, we are the first to develop a GNN-based framework for link prediction on multilayer networks which relies on structural features learned from both within-layer and across-layer link information based on different notions of overlapping multilayer neighborhoods.

## 3 PRELIMINARY DEFINITIONS

**Attributed multilayer networks.** Given a set $\mathcal{V}$ of $n$ *entities* and a set $\mathcal{L} = \{L_1, \cdots, L_\ell\}$ of *layers*, indexed in $L = \{1, \ldots, \ell\}$, with $|\mathcal{L}| = \ell \geq 2$, we denote an attributed multilayer network with $G_{\mathcal{L}} = \langle V_{\mathcal{L}}, E_{\mathcal{L}}, \mathcal{X}, \mathcal{V}, \mathcal{L} \rangle$, where $V_{\mathcal{L}} \subseteq \mathcal{V} \times L$ is the set of all entity occurrences, or *nodes*, in $\mathcal{L}$, and in particular, $V_l$ is the set of nodes in layer $l$ ($l \in L$); in the following, we might also refer to elements in $V_l$ as pairs $\langle v, l \rangle$, otherwise, i.e., if the layer is clear from the context, we will use membership notation of the form $v \in V_l$ to denote the occurrence of entity $v$ in layer $l$. $E_{\mathcal{L}}$ is the set of edges between nodes belonging to the same layer, and $E_l \subseteq V_l \times V_l$ is the set of edges in layer $l$. Each entity has a node in at least one layer, hence $\mathcal{V} = \bigcup_{l=1..\ell} V_l$, and that *inter-layer edges* exist between each node in a layer and its counterpart in a different layer. We assume independence on any relation order between the layers; if such information is available, we denote with $P(l)$ the set of valid pairings with layer $l$.

A multilayer network can be represented by a set of adjacency matrices $\mathcal{A} = \{\mathbf{A}_1, \ldots, \mathbf{A}_\ell\}$, with $\mathbf{A}_l \in \mathbb{R}^{n_l \times n_l}$ ($l \in L$), where $n_l = |V_l|$. Entities can also be associated with external information or *attributes* stored in layer-specific matrices $\mathcal{X} = \{\mathbf{X}_1, \ldots, \mathbf{X}_\ell\}$, where $\mathbf{X}_l$ is the attribute matrix for $l$; if no attributes are given for nodes in a layer, the corresponding attribute matrix is set as an identity matrix.

**Multilayer neighborhood.** For any layer $l$ and entity $v$ appearing in $l$, one basic concept characterizing the status of node $\langle v, l \rangle$ is its within-layer neighborhood locally at $l$, denoted as $\Gamma_{\langle v, l \rangle}$ and

defined as $\Gamma_{\langle v, l \rangle} = \{w \mid w \in \mathcal{V}, \ (w, v) \in E_l\}$. Since we want to take advantage of the interplay between layers in a network, we also consider neighborhoods that span across different layers to capture a notion of global connectivity of an entity. [18] propose the notion of global neighbors of an entity $v$ w.r.t. layer $l$ and $l'$, as the union of its entity-neighbors in both layers:

$$\Gamma^{(g)}_{\langle v, l, l' \rangle} = \{w \mid w \in \Gamma_{\langle v, l \rangle} \ \cup \Gamma_{\langle v, l' \rangle}\}. \tag{1}$$

For the link prediction problem, leveraging pairwise information is crucial. To capture the direct connections of two nodes in different layers, we define the *overlapping across-layer neighborhood* (OAN) of two nodes $\langle v, l \rangle$ and $\langle u, l' \rangle$ as the set of their shared entity-neighbors across layers $l, l'$:

$$\Gamma^{(oan)}_{(\langle v, l \rangle, \langle u, l' \rangle)} = \{w \mid w \in \Gamma^{(g)}_{\langle v, l, l' \rangle} \cap \Gamma^{(g)}_{\langle u, l, l' \rangle}\}. \tag{2}$$

One alternative to the above measure corresponds to the *multi-layer Adamic-Adar neighborhood* (MAAN) introduced in [2], and here denoted as $\Gamma^{(maan)}$, which considers triadic closure relations from two layers:

$$\Gamma^{(maan)}_{(\langle v, l \rangle, \langle u, l' \rangle)} = \{w \mid w \in \Gamma_{\langle v, l \rangle} \ \cap \Gamma_{\langle u, l' \rangle}\}. \tag{3}$$

Note that the above identifies an overlap between the set of neighbors of a node from one layer and the set of neighbors of another node from a different layer (e.g., mutual friends in different layers that are not necessarily known to each other). It is hence more restrictive than Eq. 2 which extends a node's neighborhood to more layers, thus allowing for identifying across-layer shared neighbors.

To enable a holistic view of the multilayer neighborhood, we can assume that a set $\mathcal{T}$ of alternative types of multilayer neighborhood measures are available. In the following, we shall consider $\mathcal{T} = \{\Gamma^{(oan)}, \Gamma^{(maan)}\}$, as two different *contexts* of multilayer structural information for nodes in our multilayer link prediction setting.

**Problem statement (Multilayer Link Prediction).** Given an (attributed) multilayer network $G_{\mathcal{L}}$, the multilayer link prediction problem is to estimate the probability of existence of edges in an arbitrary set of layers. More specifically, *for each* layer $l \in L$, the goal is to learn a function $s : V_l \times V_l \mapsto [0, 1]$ expressing the likelihood of linkage between any pair of nodes in $l$, based on their *multilayer neighborhood* information available within $l$ as well as outside $l$.

We emphasize that the problem we consider focuses on performing link prediction over all the layers of the network simultaneously, i.e., all layers in $\mathcal{L}$ are considered as target layers.

**Why within-layer *and* across-layer structural features for link prediction?** In a network-of-networks system, relying solely on within-layer information while discarding across-layer information is clearly an ineffective approach to several tasks, including link prediction [6, 25, 32, 33]. Keeping this in mind, our approach learns node structural features to compute overlapping neighborhoods between pairs of nodes, both from a within-layer and an across-layer perspective. This triggers a twofold effect: on the one hand, considering structural features of the overlapping neighborhoods between a pair of nodes *within* a layer, allows for the generalization of pairwise topological heuristics (e.g., [26]), thus enabling the exploitation of layer-specific key structural information regarding

links [42]; on the other hand, considering the structural features of the overlapping multilayer neighborhood between a pair of nodes *across* different layers, allows for the generalization of multilayer link prediction heuristics (e.g., [2]) and the exploitation of multilayer interactions that are crucial for link prediction.

Therefore, our approach to combining within- and across-layer structural information regarding links based on the overlapping multilayer neighborhoods can lead to unprecedented exploitation of multiple different structural aspects of the multilayer network. This is fundamental for a task such as link prediction, where taking advantage of any available structural information is beneficial [49]. We also argue that the joint exploitation of different types of overlapping multilayer neighborhood can facilitate the link prediction task. In the ***Appendix***, we will further motivate this aspect.

## 4 THE ML-LINK FRAMEWORK

To address the link prediction problem for (attributed) multilayer networks, we define a learning framework based on neural network models, named ML-Link.

Extracting structural information in the multilayer setting is more challenging than in the single-layer case, because we need to handle different layers of connectivity and the complex relationships arising from the multilayer structure (e.g., neighborhoods) at once. To address these challenges, our framework learns layer-tailored structural features and relies on different overlapping multilayer neighborhoods for all paired layers. This allows us to extract meaningful multilayer structural information and leverage it in an adaptive manner through an attention mechanism. Another key challenge is the ability to handle layer-specific node external features. We address this challenge by delegating it to the GNN module, which is designed for the multilayer setting.

**Overview.** Figure 1 illustrates the conceptual architecture of the proposed ML-Link, which is designed as an end-to-end trainable framework based on two components, named *NN-based node-pair neighborhood feature extraction* (NN-NPN) and *GNN-based node embeddings* (GNN-NE). The former learns node-pair-level structural features exploiting the shared neighborhood of any two nodes in different layers. The latter leverages message passing neural networks [16] for learning dense representations of nodes in the network, also exploiting available external information of nodes.

The outputs of the NN-NPN and GNN-NE components are two scores of link existence, denoted as $s_{npn}(v, u, l) \in \mathbb{R}$ and $s_{ne}(v, u, l) \in \mathbb{R}$, respectively, for any pair of nodes $v, u$ and layer $l$. These scores are then summed up to finally compute the probability $p$ of link existence between $v$ and $u$ in layer $l$, which is defined as follows:

$$p(v, u, l) = \lambda \, \sigma(s_{npn}(v, u, l)) + (1 - \lambda) \, \sigma(s_{ne}(v, u, l)), \quad (4)$$

where $\lambda$ is a learnable parameter, and $\sigma(\cdot)$ is the sigmoid function.

### 4.1 NN-based Node-pair Neighborhood Feature Extraction

The NN-NPN component is comprised of three modules that cooperate to learn the link existence scoring function $s_{npn} \colon \mathcal{V} \times \mathcal{V} \times L \mapsto \mathbb{R}$, namely: (i) *internal structure learning* (ISL) for extracting within-layer node-pair structural features, (ii) *external structure learning* (ESL) for extracting across-layer node-pair structural features, and

(iii) *context-level attention* (CLA) for adaptively weighting the importance of the information yielded from each type of multilayer neighborhood. The final score produced by the NN-NPN component for any pair of nodes $v, u$ in a layer $l$ is as follows:

$$s_{npn}(v, u, l) = (1 - \psi) s_{npn}^{\Leftrightarrow}(v, u, l) + \psi s_{npn}^{\Updownarrow}(v, u, l), \quad (5)$$

where $s_{npn}^{\Leftrightarrow}(v, u, l)$ is the score produced by ISL w.r.t. layer $l$, $s_{npn}^{\Updownarrow}(v, u, l)$ is the score produced by modules ESL and CLA, containing across-layer information, i.e., overlapping multilayer neighborhoods, and $\psi$ is a tunable parameter.

**Internal structure learning.** The ISL module relies on within-layer topological overlap for extracting features of node-pairs. To this purpose, structural node features are first extracted from the adjacency matrix of each layer via layer-specific learnable functions, based on the message passing paradigm [41]. Given a node $\langle v, l \rangle$, its structural feature $\hat{h}_{\langle v, l \rangle} \in \mathbb{R}$ is learned as follows:

$$\hat{h}_{\langle v, l \rangle} = g_2^{(l)} \Big( \sum_{w \in \Gamma_{\langle v, l \rangle}} g_1^{(l)}(\mathbf{A}_l[v, w]) \Big), \quad (6)$$

where $g_1^{(l)}, g_2^{(l)}$ are layer-specific MLPs, and $\mathbf{A}_l[v, w]$ denotes the entry in row $v$ and column $w$ of $\mathbf{A}_l$.

Next, a similarity score between any two nodes is computed by leveraging the structural features of their common neighbors. Let $\hat{\mathbf{H}}_l = \|_{v \in V_l} \hat{h}_{\langle v, l \rangle}$ be the tensor of shape $(n_l, 1)$ obtained by stacking node features $\hat{h}_{\langle v, l \rangle}$, and let $\tilde{\mathbf{H}}_l = diag(\hat{\mathbf{H}}_l) \in \mathbb{R}^{n_l \times n_l}$ be its diagonalization. The latter can be used to define the following matrix to incorporate nodes' neighborhood structural information:

$$\mathbf{Z}_l = \mathbf{A}_l \tilde{\mathbf{H}}_l. \quad (7)$$

Above, $\mathbf{Z}_l$ is in fact the matrix of structural node representations whose $v$-th row, $\mathbf{z}_{\langle v, l \rangle} \in \mathbb{R}^n$, is the node representation vector for node $\langle v, l \rangle$, where the $w$-th entry $\mathbf{z}_{\langle v, l \rangle}[w]$ is equal to $\hat{h}_{\langle w, l \rangle}$, if $w \in \Gamma_{\langle v, l \rangle}$, and 0 otherwise.

For any pair of nodes $v, u$ in layer $l$, $s_{npn}^{\Leftrightarrow}(v, u, l)$ is computed as the cosine similarity applied to vectors $z_{\langle v, l \rangle}$ and $z_{\langle u, l \rangle}$, as defined in Eq. 8:

$$
\begin{aligned}
s_{npn}^{\Leftrightarrow}(v, u, l) &= \frac{(\mathbf{z}_{\langle v, l \rangle})^{\mathrm{T}} \mathbf{z}_{\langle u, l \rangle}}{\|\mathbf{z}_{\langle v, l \rangle}\|_2 \, \|\mathbf{z}_{\langle u, l \rangle}\|_2} = \\
&= \sum_{w \in (\Gamma_{\langle v, l \rangle} \cap \Gamma_{\langle u, l \rangle})} \frac{\hat{h}_{\langle w, l \rangle}^2}{\|\mathbf{z}_{\langle v, l \rangle}\|_2 \, \|\mathbf{z}_{\langle u, l \rangle}\|_2}.
\end{aligned}
\quad (8)
$$

**External structure learning.** The ESL module is designed to capture multilayer interactions between the entities related to a target pair of nodes, using their overlapping multilayer neighborhood. Given a pair of nodes $v, u$ in layer $l$, and a set of multilayer neighborhood types $\mathcal{T}$, the goal is to compute an across-layer link existence score w.r.t. each pair of layers in $(l, l') \in l \times P(l)$ according to each context $\tau \in \mathcal{T}$. To this purpose, we compute two vectors, $\mathbf{z}_{\langle v, l \rangle}^{(\tau)}$ and $\mathbf{z}_{\langle u, l' \rangle}^{(\tau)}$, based on the shared neighbors between $\langle v, l \rangle$ and $\langle u, l' \rangle$ under $\tau$. Here, $\mathbf{z}_{\langle v, l \rangle}^{(\tau)} \in \mathbb{R}^n$ is the *context-aware vector* under $\tau$ for node $\langle v, l \rangle$, which provides a representation of node $\langle v, l \rangle$ informed of the overlapping multilayer neighborhood between $\langle v, l \rangle$ and $\langle u, l' \rangle$.

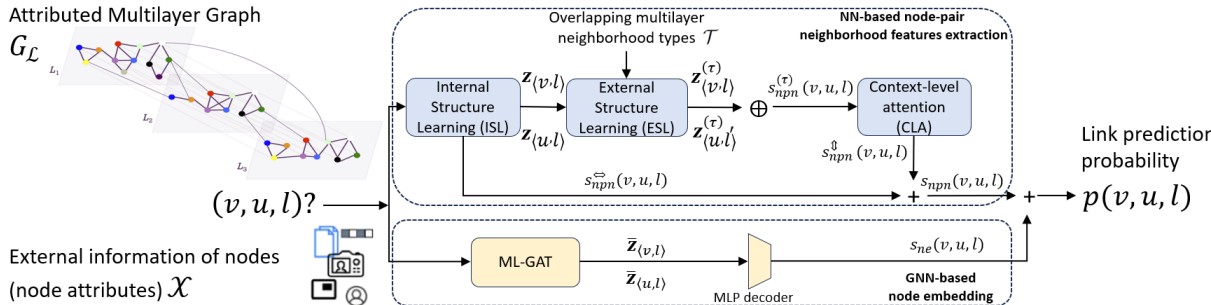

**Figure 1: Overview of our proposed ML-Link for link prediction in (attributed) multilayer networks. Blue-colored and yellow-colored modules refer to the NN-NPN and GNN-NE components, respectively.**

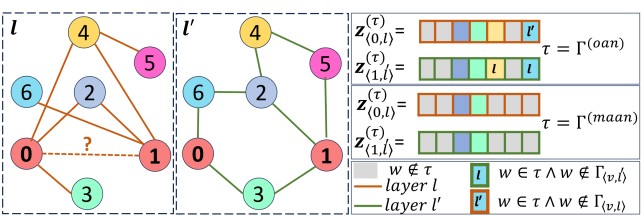

**Figure 2: Construction of the context-aware vectors for the pair of nodes 0 and 1 in layers $l$ and $l'$, resp.**

The $w$-th entry of $\mathbf{z}^{(\tau)}_{\langle v,l \rangle}$ is defined as $\hat{h}_{\langle w,l' \rangle}$, if $w \in \tau \ \wedge \ w \notin \Gamma_{\langle v,l \rangle}$, and $\hat{h}_{\langle w,l \rangle}$ otherwise. An illustrative example of construction of the context aware vectors for the pair of nodes 0 in $l$ and 1 in $l'$ is shown in Fig. 2, where non-grey-colored entries represent the overlapping neighborhoods. For the OAN context, $\Gamma^{(g)}_{\langle 0,l,l' \rangle} = \{2,3,4,6\}$, and $\Gamma^{(g)}_{\langle 1,l',l \rangle} = \{2,3,4,5,6\}$, thus $\Gamma^{(oan)} = \{2,3,4,6\}$, i.e., node 5 is not included in the resulting OAN, because it is not a neighbor of node 0 neither in layer $l$ nor in layer $l'$. Also, the 6-th entry of $\mathbf{z}^{(\tau)}_{\langle 0,l \rangle}$ corresponds to the feature learned for node $\langle 6,l' \rangle$. Similar considerations hold for the MAAN context, $\Gamma^{(maan)}$.

The ESL link existence score under $\tau$ for $v, u$ in $l$ is finally computed through the cosine similarity of the context aware node representations for each pair of comparable layers $(l, l')$, as shown in Eq. 9:

$$s^{(\tau)}_{npn}(v,u,l) = g^{(\tau)}_3 \Big( \bigoplus_{l' \in P(l)} \eta^{(\tau)} \cdot \frac{(\mathbf{z}^{(\tau)}_{\langle v,l \rangle})^{\mathrm{T}} \mathbf{z}^{(\tau)}_{\langle u,l' \rangle}}{\big\| \mathbf{z}^{(\tau)}_{\langle v,l \rangle} \big\|_2 \big\| \mathbf{z}^{(\tau)}_{\langle u,l \rangle} \big\|_2} \Big), \qquad (9)$$

where $\eta^{(\tau)}$ is a weighing coefficient associated with context $\tau$ to control the weights of each pair of layers $(l, l')$; by default $\eta^{(\tau)}$ is set to 1, however an alternative setting as $\frac{1}{k_l \cdot k_{l'}}$, with $k_l$ denoting the average degree of layer $l$, would be useful for penalizing link likelihoods more on denser layers. Moreover, $g^{(\tau)}_3$ in Eq. 9 denotes a transformation function for yielding a high level representation of the across-layer information scores; $\bigoplus$ is the aggregator operator (e.g., sum or concatenation). In our framework, $\bigoplus$ is chosen as the summation operator, and $g^{(\tau)}_3$ corresponds to a MLP.

Note that, unlike [42], the link predictions are produced using the cosine similarity function, rather than the dot product, as the

normalization factor reveals to be beneficial for the learning process. In the ***Appendix***, we give empirical evidence of the effectiveness of our choice.

**Context-level attention.** Given the different predictive information obtaied w.r.t. the various contexts (i.e., overlapping multilayer neighborhood types) $\tau$, we introduce a *self-attention mechanism* to weigh the importance that each context has in predicting link $(v,u)$ in layer $l$. The overall likelihood computed by the node-pair neighborhood feature extraction module is hence defined as:

$$s^{\Updownarrow}_{npn}(v,u,l) = \sum_{\tau \in \mathcal{T}} \alpha^{(\tau,l)} s^{(\tau)}_{npn}(v,u,l), \qquad (10)$$

where $\alpha^{(\tau,l)}$ is the normalized amount of attention for neighborhood of type $\tau$ in layer $l$. To learn the attention coefficients, we follow the formulation used in [40]:

$$\alpha^{(\tau,l)} = \frac{\exp\left(m^{(\tau,l)}\right)}{\sum_{\tau \in \mathcal{T}} \sum_{l \in L} \exp\left(m^{(\tau,l)}\right)} \quad \text{with,}$$

$$m^{(\tau,l)} = \frac{1}{|E_l|} \sum_{(u,v) \in E_l} \mathbf{a}^{\mathrm{T}} g_{att}(s^{(\tau)}_{npn}(v,u,l)), \qquad (11)$$

where $\mathbf{a}$ is the trainable attention vector, and $g_{att}$ is a MLP.

Note that, to gain insights into which contexts the CLA module focused on through the conducted experiments, we provide a visual analysis of the learned attention coefficients, in the ***Appendix***.

## 4.2 GNN-based Node Embedding

The GNN-NE component is initially in charge of learning node-level dense representations (embeddings) $\bar{\mathbf{Z}}$, possibly incorporating layer-specific nodes' attributes $(\mathcal{X})$, based on intra- and inter-layer topology information in a unified way:

$$\bar{\mathbf{Z}} = F_\theta(G_{\mathcal{L}}, \mathcal{X}). \qquad (12)$$

$\bar{\mathbf{Z}}$ is the learned node-embedding matrix with shape $(|\mathcal{V}_{\mathcal{L}}|, d)$, so that $\bar{\mathbf{z}}_{\langle v,l \rangle}$ is the embedding for node $\langle v,l \rangle$. Moreover, $F_\theta$ denotes a GNN model, with learnable parameters $\Theta$, specifically dealing with multilayer networks. In this work, we resort to the ML-GNN framework proposed in [43], which generalizes the message passing paradigm for multilayer graphs. Our choice for ML-GNN over existing GNN-based alternatives (e.g., [37]) is mainly motivated

since ML-GNN, has shown to be particularly effective in aggregating topological neighborhood information from different layers directly into the propagation rule of the GNN component, i.e., during its forward learning phase, in order to make the embedding of an entity in a particular layer depending on both its neighbors in that layer (i.e., within-layer neighborhood) and on its neighbors located in other layers where the entity occurs (i.e., outside-layer neighborhood). Moreover, ML-GNN architecture is versatile w.r.t. both convolutional and attentive GNN models. In our experimental evaluation of ML-Link, we shall refer to instantiation with the GATv2 architecture, based on a self-attention mechanism, hereinafter referred to as ML-GAT [7]. Please note that testing the impact of further alternative GNNs on our framework is beyond the scope of this work.

To predict the final GNN-NE function, $s_{ne}$, the embeddings learned by the ML-GAT model are pairwise used as input of layer-specific MLP decoders based on the Hadamard product, as shown in Eq. 13:

$$s_{ne}(v, u, l) = g_4^{(l)}(\bar{z}_{\langle v, l \rangle} \circ \bar{z}_{\langle u, l \rangle}), \tag{13}$$

where $\circ$ is the element-wise product, and $g_4^{(l)}$ is the MLP predictor that outputs the GNN-NE likelihood of link formation in $l$.

We point out that, due to the GNN-NE component, our ML-Link can normally work in case no overlapping neighbors exist between a pair of nodes, due to the ability of ML-Link to adaptively combine information from the two components (cf. Eq. 4).

### 4.3 Loss Function

The loss function of ML-Link is defined for a binary classification task, where existing edges in the input multilayer network are treated as positive examples, and a certain amount of non-linked node-pairs are treated as negative examples. Given any pair of nodes $v, u$ in layer $l$, let us denote with $y_{(v,u)}^{(l)}$ the associated ground-truth value, i.e., 1 if the nodes are linked to each other, 0 otherwise.

We optimize the binary cross entropy of each of the learned link-existence scoring functions, i.e., $s_{npn}$, $s_{ne}$, and $p$. Considering the latter, the loss function associated with the overall link-existence scores is:

$$\mathcal{I}_p = \sum_{l \in L} \sum_{(v,u) \in E_l^{(train)}} \mathcal{B}(p(v, u, l), y_{(v,u)}^{(l)}), \tag{14}$$

where $\mathcal{B}$ is the binary cross entropy function, and $E_l^{(train)}$ is the set of (positive and negative) training node-pairs in $l$.

Note that $\mathcal{I}_p$ is mainly in charge of balancing, through the learnable parameter $\lambda$, the contributions of the scores yielded by the NN-NPN and GNN-NE components, where the latter independently learn their prediction scores. Analogously, we apply binary cross entropy to the $s_{ne}$ and $s_{npn}$ scores, respectively, to compute their corresponding $\mathcal{I}_{ne}$ and $\mathcal{I}_{npn}$ loss functions. Finally, the overall loss function $\mathcal{I}$ is computed by summing up $\mathcal{I}_{npn}$, $\mathcal{I}_{ne}$, and $\mathcal{I}_p$.

Our choice of combining the individual losses, rather than relying on $\mathcal{I}_p$ only, is motivated since this is likely to make the training process more stable; intuitively, on the one hand, during the optimization of each component's loss, wrong scores can be directly detected during the training, and on the other hand, by optimizing $\mathcal{I}_p$, errors in individual component could get overlooked if they compensate each other.

We emphasize that ML-Link is trained in end-to-end manner to jointly learn the link existence score between any pair of nodes in all the layers of the network. Therefore, *it does not require a separate training stage for each layer*, but one single training for all layers altogether (cf. Eq. 14).

## 5 EXPERIMENTAL EVALUATION

**Evaluation goals.** We design our experimental evaluation to pursue the following objectives: (1) to measure and compare the effectiveness of ML-Link w.r.t. 11 machine-learning-based competitors and 6 link-prediction heuristics; (2) to carry out an ablation analysis to show the impact of each constituting module of ML-Link on the link prediction task; (3) to assess the sensitivity of ML-Link w.r.t. its main hyper-parameter, i.e., $\psi$; (4) to test ML-Link scalability capabilities. Also, in the *Appendix*, we discuss computational complexity aspects of ML-Link.

**Data.** We used publicly available real-world multilayer networks for our main experiments. In addition, we built synthetic networks of varying size, based on the Watts-Strogatz generative model, for studying the efficiency of ML-Link. A description of the real-world networks, the generation process of the synthetic networks, as well as their structural characteristics, are reported in the *Appendix*.

**Competing methods.** We compare the proposed ML-Link to *an ensemble of traditional heuristic algorithms*, which include Common Neighbors, Preferential Attachment, Adamic-Adar, Jaccard, Resource allocation index, and SimRank (cf. *Appendix*). We also consider the following methods for link prediction in multilayer graphs: MAGMA [14], Pujari [36], Jalili [19], Hristova [18], MAA [2], MELL [30], CrossMNA [12], ML-GAT [43] and GATNE [9]. Yet, we include widely used methods for link prediction on single layer networks, such as Neo-GNN [42] and SEAL [45].

**Experimental setting.** Given a multilayer network, we performed link prediction *on all its layers*, at the same time; also, we refer to a transductive setting, i.e., all the nodes of a network are available at training time. For the evaluation, we considered the whole set of edges of a multilayer network, split it into training, test and validation edge-sets using 10-fold cross validation, and projected the edges of each fold onto the layers; for instance, if the edge $(u, v)$ was in the current training/test/validation split, we took it in the training/test/validation split of layer $l$ only if it appeared in layer $l$. The negative training/test/validation non-linked node-pairs were randomly sampled for each layer, in the same amount of the training/test/validation positive edges. We measured the *area under the ROC curve* (AUC) and the *average precision* (AP) scores on the union of layer-specific test sets of linked/non-linked node-pairs.

For the single-layer methods, Neo-GNN and SEAL, we first trained a separate model on each layer of the network, then we performed inference on the test set of each layer to fit single-layer approaches for the training and the evaluation on multilayer networks. Finally, we computed the AUC and AP values on the concatenation of the link existence scores provided by each model. In the *Appendix*, we describe the setting of the hyperparameters for each method.

**Table 1: Comparative evaluation: AUC (top) and AP (bottom) values on real multilayer networks. Bold, resp. underlined, values correspond to the best, resp. second-best, scores on each network. OOT: Out-of-Time, OOM: Out-of-Memory**

| Method | Cs-Aarhus | CKM | Elegans | Lazega | DkPol | ArXiv |
|---|---|---|---|---|---|---|
| ML-Link | **97.208** | **99.269** | **99.646** | **99.557** | **99.552** | **99.342** |
|  | **97.348** | **99.268** | **99.645** | **99.579** | **99.515** | **99.470** |
| Ensemble | 89.831 | 73.528 | 80.322 | 81.860 | 92.124 | _99.171_ |
|  | 89.520 | 72.906 | 79.759 | 80.398 | 92.423 | _99.293_ |
| MAGMA [14] | 85.606 | _92.341_ | 96.176 | _82.188_ | 90.749 | 96.238 |
|  | 80.619 | _89.659_ | _96.335_ | 79.036 | 89.632 | 96.114 |
| Pujari [36] | 83.218 | 69.225 | 77.017 | 64.564 | 79.241 | OOT |
|  | 75.559 | 74.774 | 76.763 | 58.747 | 71.735 | OOT |
| Jalili [19] | 80.717 | 79.730 | 67.987 | 59.801 | 73.408 | OOT |
|  | 76.270 | 70.188 | 65.248 | 55.223 | 72.701 | OOT |
| Hristova [18] | 79.766 | 71.803 | 56.198 | 55.054 | 62.586 | OOT |
|  | 60.176 | 61.44 | 54.097 | 53.626 | 53.295 | OOT |
| MAA [2] | _92.083_ | 85.151 | 86.025 | 79.682 | 90.719 | OOT |
|  | _91.611_ | 86.692 | 84.422 | 78.260 | 89.438 | OOT |
| MELL [30] | 73.641 | 68.357 | 82.093 | 64.262 | 45.918 | OOM |
|  | 77.517 | 77.521 | 88.644 | 70.328 | 48.570 | OOM |
| CrossMNA [12] | 78.589 | 88.317 | 88.389 | 74.54 | 68.371 | 98.318 |
|  | 75.457 | 87.859 | 87.203 | 69.68 | 61.268 | 98.426 |
| ML-GAT [43] | 89.432 | 88.517 | _96.307_ | 72.623 | 85.382 | 82.635 |
|  | 88.754 | 86.751 | 95.236 | 69.047 | 84.015 | 76.617 |
| GATNE [9] | 85.096 | 90.033 | 88.389 | 78.352 | 75.579 | 98.914 |
|  | 84.459 | 88.445 | 87.203 | 75.231 | 73.317 | 99.187 |
| Neo-GNN [42] | 83.370 | 89.094 | 82.793 | 78.956 | 81.084 | 92.176 |
|  | 82.986 | 87.591 | 82.405 | 78.428 | 81.983 | 93.847 |
| SEAL [45] | 81.986 | 83.898 | 87.979 | 81.429 | _95.004_ | 98.823 |
|  | 82.316 | 83.651 | 86.517 | _80.140_ | _94.684_ | 98.816 |

## 6 RESULTS

**Comparative evaluation.** Table 1 shows the AUC and the AP values obtained on real-world multilayer networks by ML-Link, competing methods, and baselines. Note that, for the sake of presentation, we summarize in row *Ensemble* the scores by the best-performing baseline on each particular dataset (full details are reported in the ***Appendix***). Also, ML-Link results correspond to the use of all types of multilayer context ($\mathcal{T}$), which reveals to be the best setting as we shall discuss next in the ablation study.

Looking at Table 1, several remarks stand out. First of all, ML-Link outperforms all the other methods in all cases, yielding the best average results across all datasets (on average, 99.09 AUC), followed by MAGMA (with percentage decrease of about 10% on the averaged results), MAA and SEAL. ML-Link consistently outperforms Neo-GNN, which is the only other neural method leveraging overlapping neighborhoods; this supports our initial intuition that our approach of considering both within- and across-layer structural features improves the link prediction performance.

As expected, the two best-performing competitors, i.e., MAGMA and MAA, are all multilayer methods. The GNN-based approaches, GATNE and ML-GAT, are among the best multilayer methods; however, both achieve low performance scores on Lazega. Note that this is a network with high average degree (12.148 layer average), high clustering coefficient (0.351), low diameter (5.667), low average path length (2.211) and present multiple hub nodes. Also, the same nodes across different layers share similar degree centrality scores. Given the above structural properties of the network, ML-GAT

and GATNE might learn similar hidden representations for several nodes, thus complicating the task of distinguishing between existing and non-existing link.[1] By contrast, our ML-Link learns distinct structural features for each node at each layer, and can extract complex patterns of structural information regarding links, which may help the model in discerning between links and non-links.

GATNE performs poorly also on DkPol, which has similar intra-layer characteristics w.r.t. Lazega, but with a marked unbalance between layers: the third layer is two orders of magnitude larger than the other two, and has a significantly higher average degree (79.922), higher clustering coefficient (0.520), lower diameter (4), and lower modularity (0.183). Such a lack of structural coherence in DkPol layers is also detrimental for the performance of CrossMNA and MELL, where they achieve their worst AUC and AP.

Considering other multilayer methods, Hristova yields the overall worst performance, followed by MELL, Jalili and Pujari. By contrast, as previously mentioned, MAA (which uses the MAAN neighborhood for computing link existence scores) and MAGMA perform generally well; however, MAGMA performance is negatively affected on Lazega, likely due to its high density that hinders MAGMA to learn meaningful patterns.

Both the single-layer GNN methods, i.e., Neo-GNN and SEAL, and the ensemble of heuristics provide good results on most networks, sometimes even better than multilayer methods. However, note that such methods deal with each layer independently, and especially the heuristics appear to particularly suffer from the presence of layers that are far from a small-world model, like in CKM and Elegans.

On the ArXiv network, which exhibits a high clustering coefficient (0.650) and modularity (0.942), competitors perform well, with the exception of those models that running out of time/memory.[2] ML-GAT relatively low performance could be ascribed to the network's high dimensionality and clustering coefficient of each layer, where nodes within a cluster might negatively impact on the ML-GAT attentive-based ability to assign diverse weights to each edge.

Overall, empirical evidence has demonstrated the high effectiveness of our ML-Link on networks with different structural properties. This is ascribed to its ability to extract complex within-layer and across-layer structural patterns regarding link formation. Next, in the ablation analysis, we show that when across-layer information are integrated and adaptively combined, there is a a performance improvement w.r.t. the case where only within-layer information are used, by a large margin.

**Ablation analysis.** To validate the architectural design of our ML-Link, we assessed the effectiveness of each component, by examining several simplified variants of ML-Link.

Table 2 summarizes the performance results obtained by ML-Link and its simplified variants on a subset of our evaluation networks. As expected, the full framework is the best-performing version,

---

[1] ML-GAT follows a message passing scheme in which the embedding of a node in one layer is computed by considering all its neighbors in all the layers, which can lead to over-smoothing problems [34, 43] due to the excessive aggregation of information. Similarly, GATNE learns node representations in one layer and a base embedding that is shared for the same nodes across different layers; also, its optimization procedure employs a meta-path based random walks strategy combined with the skip-gram model, ensuring that nodes appearing in the same context have similar embeddings.
[2] Hristova, Jalili and Pujari ran out-of-time (i.e., > 24h), likely due to their NetworkX implementation, while MELL faced out-of-memory issues.

**Table 2: Ablation study: AUC (top) and AP (bottom) values of ML-Link and its simplified versions. Bold and underlined values correspond to the best and second-best scores, resp.**

| Method | Cs-Aarhus | CKM | Elegans | Lazega | DkPol |
|---|---|---|---|---|---|
| GNN-NE | 89.432 | 88.517 | 96.307 | 72.603 | 85.382 |
| | 88.754 | 86.751 | 95.236 | 69.047 | 84.015 |
| ISL | 84.622 | 62.450 | 75.158 | 78.905 | 85.924 |
| | 84.848 | 68.512 | 73.560 | 77.035 | 85.699 |
| ISL w/ GNN-NE | 91.16 | 89.341 | 96.253 | 80.577 | 90.398 |
| | 91.07 | 88.645 | 95.377 | 79.867 | 92.031 |
| ISL w/ ESL ( $\Gamma^{(oan)}$ ) | 90.410 | 72.301 | 82.172 | 78.294 | 85.531 |
| | 89.946 | 76.821 | 79.523 | 77.455 | 82.478 |
| ISL w/ ESL ( $\Gamma^{(maan)}$ ) | 89.175 | 70.723 | 79.369 | 80.527 | 87.915 |
| | 88.769 | 75.393 | 77.826 | 78.456 | 86.236 |
| ISL w/ ESL | 90.284 | 73.191 | 82.693 | 81.254 | 87.521 |
| | 89.566 | 77.709 | 80.294 | 79.200 | 86.228 |
| NN-NPN | 95.927 | 98.561 | 98.828 | 99.023 | 98.036 |
| | 95.481 | 97.576 | 98.865 | 99.104 | 98.560 |
| ML-Link | **97.208** | **99.269** | **99.646** | **99.557** | **99.552** |
| | **97.348** | **99.268** | **99.645** | **99.579** | **99.515** |

while the worst is ISL, whose AUC and AP scores in most datasets are comparable to Pujari in Table 1. As the overlapping multilayer neighborhoods are gradually integrated within the ESL module, the performance tends to significantly increase. Nonetheless, when both contexts are considered without being adaptively combined (ISL w/ ESL), the performance improvement is marginal compared to cases where only one single context is employed (i.e., ISL w/ ESL ($\Gamma^{(oan)}$), and ISL w/ ESL ($\Gamma^{(maan)}$). On the other hand, when the CLA module is used, NN-NPN outperforms all competing methods shown in Table 1, achieving similar performance as ML-Link. This is due to the attention mechanism, enabling each layer to learn the importance to be given for each type of overlapping multilayer neighborhood. Regarding the GNN-NE component, it stands out that it is also beneficial for the performance. This is particularly evident considering that, despite we use the identity matrix initialization, the ISL w/ GNN-NE version achieves an average AUC percentage increase of about 15% w.r.t. the ISL version. The above results demonstrate the importance of each module of ML-Link, and that all are needed to maximize performance.

**Sensitivity analysis.** A further stage of evaluation concerned the impact of the $\psi$ hyper-parameter, which weighs the importance of ISL vs. modules ESL and CLA (cf. Eq. 5). To avoid any bias from the GNN-NE component, we discarded it in the evaluation; experiments with the full framework are described in the ***Appendix***.

Figure 3 shows AUC and AP results by varying $\psi$ from 0 to 1; note that $\psi = 0$ discards the overlapping multilayer neighborhood contributions (i.e., ESL module is off), which is conversely the only information used when $\psi = 1$. In the figure, it can be noticed a certain robustness of ML-Link w.r.t. $\psi$, as we consistently observe on all networks that the performances are higher and relatively stable when $\psi \in [0.2, 0.9]$, with peaks reached within [0.5, 0.8]. Note that when $\psi = 1$, the AUC values tend to slightly decrease, but do not degrade as observed when $\psi = 0$, which is due to the overlapping multilayer neighborhood contributions that encompass the within-layer connectivity information.

**Efficiency analysis.** Table 3 compares the training time of our framework with that of the strongest competing method, i.e., MAGMA,

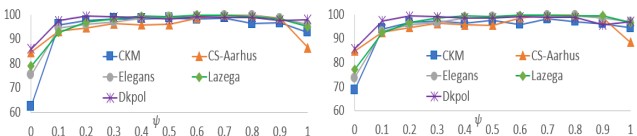

**Figure 3: AUC (left) and AP (right) values by varying $\psi$.**

**Table 3: Training time (min) of ML-Link vs. MAGMA.**

| | $\beta = 0.1$ | | | $\beta = 0.5$ | | |
|---|---|---|---|---|---|---|
| | ML-Link | | MAGMA | ML-Link | | MAGMA |
| $|V_{\mathcal{L}}|$ | GPU | CPU | CPU | GPU | CPU | CPU |
| 1500 | 0.013 | 0.095 | 0.027 | 0.023 | 0.167 | 0.043 |
| 3000 | 0.020 | 0.403 | 0.154 | 0.034 | 0.718 | 0.453 |
| 4500 | 0.030 | 1.193 | 0.670 | 0.059 | 2.271 | 1.595 |
| 6000 | 0.060 | 3.792 | 5.173 | 0.120 | 7.280 | 5.636 |
| 7500 | 0.102 | 7.938 | 8.489 | 0.211 | 15.589 | 12.478 |
| 9000 | 0.179 | 17.627 | 20.339 | 0.397 | 33.054 | 29.131 |
| 10500 | 0.339 | 31.074 | 34.781 | 0.740 | 60.217 | 54.932 |
| 12000 | 0.641 | 62.395 | 77.394 | 1.322 | 117.877 | 148.338 |
| 13500 | 1.366 | 88.707 | 127.924 | 2.659 | 184.375 | 235.066 |

on two sets of small-world networks synthetically generated with rewiring probability $\beta$ set to 0.1 and 0.5, and by varying the number of nodes from 500 to 13500 on each set. To make the comparison as much fair as possible, we took two actions: (i) since the available code of MAGMA is for CPU only, we show the training time of ML-Link on both CPU and GPU, and (ii) we used the suboptimal ISL w/ GNN-NE version of ML-Link to achieve comparable AUC results with MAGMA (cf. Tables 1–2).

Considering the CPU times, we observe that as the number of nodes increases, ML-Link tends to be faster than MAGMA for both sets of networks. On networks corresponding to $\beta = 0.5$, both GPU and CPU times of ML-Link are approximately doubled compared to the case when $\beta = 0.1$; this is explained since we needed to double the number of epochs (20) to achieve comparable performance with MAGMA. The latter also shows slower performance when $\beta = 0.5$, as this results in a higher degree of randomness, making the rule extraction process by MAGMA more challenging.

## 7 CONCLUSIONS

We presented ML-Link, a novel neural-network-based learning framework for link prediction on (attributed) multilayer networks, which jointly learns GNN-based multilayer node embeddings and NN-based node-pair structural features leveraging different types of overlapping multilayer neighborhood, thus effectively utilizing across-layer information for link estimation. Results have shown that ML-Link consistently outperforms several baselines and competing methods on different real-world multilayer networks, is faster than the most accurate competing method, and is robust to the hyper-parameter controlling the impact given to the overlapping multilayer neighborhoods.

***Reproducibility Note:*** Please refer to the Technical Appendix for further information on our approach, evaluation data and experiments. *Source code and evaluation data are made available to the research community at https://shorturl.at/cOUZ4*

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

## A NOTATIONS

Frequently used symbols throughout the main paper are summarized in Table 4.

## B MORE ON OVERLAPPING MULTILAYER NEIGHBORHOODS

Let us consider the example in Figure 4 to elaborate more on the OAN and the MAAN neighborhood types (cf. Eqs 2 and 3 in the main text). For the target pair 0, 1 in layers $l$ and $l'$, the OAN overlapping multilayer neighborhoods is $\Gamma^{(oan)}_{(\langle 0,l\rangle,\langle 1,l'\rangle)} = \{2, 3, 4, 6\}$ because it takes into account the union of the connections of 0 and 1 in both layers, that is, $\Gamma^{(g)}_{\langle 0,l,l'\rangle} = \{2, 3, 4, 6\}$ and $\Gamma^{(g)}_{\langle 1,l',l\rangle} = \{2, 3, 4, 5, 6\}$. The MAAN overlapping multilayer neighbors is $\Gamma^{(maan)}_{(\langle 0,l\rangle,\langle 1,l'\rangle)} = \{3\}$, since $\Gamma_{\langle 0,l\rangle} = \{2, 3, 4\}$ and $\Gamma_{\langle 1,l'\rangle} = \{3, 5\}$. In this example, it can be noticed its asymmetry property [2], i.e., $\Gamma^{(maan)}_{(\langle v,l\rangle,\langle u,l'\rangle)} \neq \Gamma^{(maan)}_{(\langle u,l\rangle,\langle v,l'\rangle)}$, observing that $\Gamma^{(maan)}_{(\langle 1,l\rangle,\langle 0,l'\rangle)} = \{6\}$.

As a realistic scenario of how the joint exploitation of the two types of overlapping multilayer neighborhoods can be beneficial, consider Fig. 4 as an OSN with two layers, where $l$ models friendship and $l'$ models group membership, e.g., collaboration, trust or shared interests. For predicting links in layer $l$ (friend recommendation), exploiting a high overlap in the OAN multilayer neighborhoods between nodes could be beneficial, since two users (0 and 1) might become friends if they share multiple friends (2 and 4), or people following the same groups (3), or users not necessarily in the same layer (6). Conversely, for predicting links in the second layer (group membership), the OAN overlapping neighborhood could be less beneficial, since a potentially high overlap of the latter (e.g., users sharing many friends) may not necessarily mean that people are interested to join the same groups. Relying on richer information about the pair of target users (0 and 1), such as their triadic relations when they belong to different layers (i.e., the MAAN multilayer neighbor between 0 in $l$ and 1 in $l'$) could be more meaningful, since two people might be more likely to join the same group in the future (e.g. 0 and 1 will connect in $l'$), if one of the two people has a friend (0 is connected to 3 in $l$) who is in the same group of the other person (3 is connected to 1 in $l'$).

## C COMPUTATIONAL TIME COMPLEXITY

We discuss the computational time complexity of the GNN-NE and NN-NPN components separately. We assume sparse matrices, that the size of the hidden dimensions for each neural module is $d$, with $d \ll n$, and that link prediction is performed on each layer of the multilayer graph.

The GNN-NE component takes $O(K|V_{\mathcal{L}}|d^2 + K|E_{\mathcal{L}}|d)$ [43], where $K$ is the number of GNN's hidden layers, and $d$ the size of the hidden dimension. Regarding the NN-NPN component, learning the structural node features and constructing $Z_l$ for layer $l$ takes $O(|E_l|d)$. Then, since $Z_l$ is represented as a sparse matrix, the time complexity of computing the ISL score is $O(|E_l|)$ for each layer. Thus, considering all the layers, and the ESL, we have $O(|E_{\mathcal{L}}|d) + \sum_{\tau \in \mathcal{T}} T^{(\tau)}$ where $T^{(\tau)}$, is the complexity related to the specific context $\tau$. For example, taking into account the OAN multilayer neighborhood (i.e., $\tau = \Gamma^{(oan)}$), we have that $T^{(\tau)} = \sum_{l \in L} \sum_{l' \in P(l)} O(|E_l + E_{l'}|)$

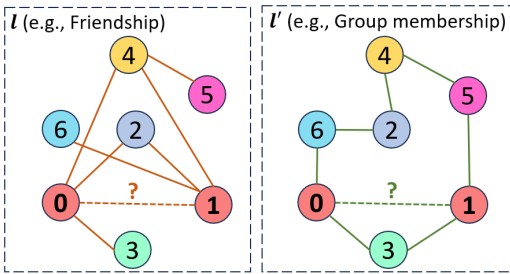

**Figure 4: Multiplex network with 2 layers, denoted with $l$ and $l'$, 7 entities and 14 edges. The target pair of nodes in both layers ($0$ and $1$) is red-colored with bold identifiers.**

due to the sum of the adjacency matrices $A_l$ and $A'_l$ and the computation of the overlapping neighborhood on the resulting matrix. Thus, assuming the worst-case scenario where each layer $l$ is paired with each other (i.e., $P(l) = L \setminus \{l\}$), the overall time complexity is $O(|E_{\mathcal{L}}|^2)$. On the contrary, if each layer is paired with a constant number of other layers, the overall time complexity can be reduced to $O(|E_{\mathcal{L}}|)$. Finally, learning the attention coefficients to weigh each $\tau$ is $O(|\mathcal{T}||E_{\mathcal{L}}|d)$.

## D DATA DESCRIPTION

### D.1 Real-world datasets

We describe the real-world multilayer networks used in our evaluation, whose main statistics are reported in Table 5.

**Cs-Aarhus** [28] is a social network with five types of (undirected) relations between employees (nodes) in the Department of Computer Science at Aarhus University. Layers corresponds to online and offline relations: Co-authorship, Facebook, Leisure, Lunch, Work relations.

**CKM** [13] is a network built from social information obtained from physicians (nodes) when adopting new drugs. It includes three directed layers that represent interactions between physicians: ask for advice, discussion and friendship.

**Lazega** [23] is a directed social network between partners and associates in a corporate law partnership, where its three layers correspond to Advice, Friendship and Co-workship relationship.

**Elegans** [11] is an undirected multiplex network representing the Caenorhabditis elegans connectome. Entities are neurons and layers correspond to different synaptic junctions: electric, chemical monadic, and polyadic.

**DkPol** (Dansk Politik) [27] is a network with three types of direct online relations (Retweet, Reply, Follow) on Twitter between Danish politicians running for the parliament who also had a Twitter account.

**ArXiv** [15] is a co-autorship (undirected) network which consists of 13 layers corresponding to different arXiv categories.

### D.2 Synthetic networks

For the efficiency analysis, we generated two sets of synthetic networks with 3 layers and a different number of entities (from 500 to 4000 with step 500) using the Watts-Strogatz model. Each layer was generated with a different realization of Watts-Strogatz,

**Table 4: Summary of notations and their description.**

| Notations | Description |
|---|---|
| $G_{\mathcal{L}}$ | An attributed multilayer network graph |
| $\mathcal{V}, V_{\mathcal{L}}, E_{\mathcal{L}}, \mathcal{X}$ | Set of entities/nodes/edges and attributes of $G_{\mathcal{L}}$ |
| $\mathcal{L}, L, \ell$ | Set of layers, set of layer indices, and number of layers |
| $l, P(l)$ | Generic layer, and set of valid pairings with layer $l$ |
| $V_l, E_l$ | Set of nodes, and edges in layer $l$ |
| $A_l, n_l$ | Adjacency matrix of layer $l$, number of nodes in layer $l$ |
| $v, \langle v, l \rangle$ | Entity $v$, and the associated node (or instance) in layer $l$ |
| $A_l[v, w]$ | $v, w$-th entry of the adjacency matrix $A_l$ |
| $\langle v, l \rangle, \langle u, l' \rangle$ | Pair of nodes in different layers |
| $\Gamma_{\langle v, l \rangle}$ | Local neighborhood in layer $l$ for node $\langle v, l \rangle$ |
| $\Gamma_{\langle v, l, l' \rangle}^{(g)}$ | Global neighborhood of $v$ w.r.t. layer $l$ and $l'$ |
| $\mathcal{T}, \tau$ | Set of multilayer contexts, and a generic multilayer context |
| $\Gamma_{(\langle v, l \rangle, \langle u, l' \rangle)}^{(oan)}, \Gamma_{(\langle v, l \rangle, \langle u, l' \rangle)}^{(maan)}$ | OAN and MAAN multilayer neighborhoods between nodes $\langle v, l \rangle$ and $\langle u, l' \rangle$ |
| $\hat{h}_{\langle v, l \rangle}$ | Structural feature for node $\langle v, l \rangle$ |
| $\eta^{(\tau)}$ | Weighting factor associated with $\tau$ |
| $\alpha^{(\tau, l)}$ | Attention coefficient learned for $\tau$ and layer $l$ |
| $Z_l$ | Structural representations matrix for layer $l$ |
| $\mathbf{z}_{\langle v, l \rangle}, \mathbf{z}_{\langle v, l \rangle}[w]$ | Structural representation of node $\langle v, l \rangle$, and its $w$-th component, resp. |
| $\mathbf{z}_{\langle v, l \rangle}^{(\tau)}, \mathbf{z}_{\langle v, l \rangle}^{(\tau)}[w]$ | Context aware representation under $\tau$ for node $\langle v, l \rangle$, and its $w$-th component, resp. |
| $\bar{Z}$ | Node embedding matrix |
| $\bar{\mathbf{z}}_{\langle v, l \rangle}$ | Node embedding for node $\langle v, l \rangle$ |
| $g_1^{(l)}, g_2^{(l)}, g_4^{(l)}$ | MLPs for layer $l$ |
| $g_{att}$ | Attention MLP |
| $\bigoplus, g_3^{(\tau)}$ | Generic aggregation operator and transformation function for context $\tau$, resp. |
| $s_{npn}^{(\tau)}(v, u, l)$ | ESL link existence score associated with $\tau$ |
| $s_{npn}^{\Leftrightarrow}(v, u, l), s_{npn}^{\Updownarrow}(v, u, l)$ | ISL and across-layer link existence score for $(v, u)$ in layer $l$ |
| $p(v, u, l), s_{npn}(v, u, l), s_{ne}(v, u, l)$ | Overall, NN-NPN and GNN-NE link existence scores for $(v, u)$ in layer $l$ |

and using the 0.5% of the number of entities as average degree for obtaining similar networks to the real ones. Table 6 reports the basic statistics of the synthetically generated networks.

# E  ADDITIONAL DETAILS ON EXPERIMENTAL METHODOLOGY

## E.1  Baselines

In our experimental evaluation, baselines correspond to local and global heuristic algorithms for single-layer networks. In the following, we will use $\Gamma_v$ for denoting the neighborhood of node $v$ in a single-layer graph.

**Common Neighbors (CN)** [26] computes the likelihood of connection between two nodes by counting the number of shared neighbors:

$$CN(v, u) = |\Gamma_v \cap \Gamma_u|. \tag{15}$$

CN is particularly used in the domain of social network for friend recommendation. Indeed, it has been shown that there is a correlation between the number of shared neighbors and the likelihood of linkage between two nodes [31].

**Jaccard similarity.** Similarly to CN, Jaccard index is also based on the shared neighbors between two nodes:

$$J(v, u) = \frac{|\Gamma_v \cap \Gamma_u|}{|\Gamma_v \cup \Gamma_u|}. \tag{16}$$

It normalizes the CN score, quantifying the probability of selecting the common neighbors between two nodes given all the neighbors of either nodes.

**Adamic-Adar.** [1] quantifies the similarity between two nodes by assigning weights to common neighbors:

$$AA(v, u) = \sum_{w \in \Gamma_v \cap \Gamma_u} \frac{1}{\log |\Gamma_w|}, \tag{17}$$

**Table 5: Main structural characteristics of the real-world networks; deg, apl, c, den and diam correspond to average degree, average path length, clustering coefficient, density and diameter, resp. Mod. and comm, correspond to the modularity value and the number of communities obtained with Louvain [5] algorithm, with resolution equal to 1.**

| Network | $\|\mathcal{V}\|$ | $\|V_{\mathcal{L}}\|$ | $\|E_{\mathcal{L}}\|$ | $\ell$ | deg. | apl. | cc. | den | diam. | mod. | comm. |
|---|---|---|---|---|---|---|---|---|---|---|---|
| Cs-Aarhus | 61 | 224 | 620 | 5 | 1.680 | 1.667 | 0.429 | 0.073 | 8 | 0.757 | 8 |
| | | | | | 7.750 | 1.956 | 0.481 | 0.250 | 4 | 0.332 | 4 |
| | | | | | 3.745 | 3.123 | 0.343 | 0.081 | 8 | 0.571 | 6 |
| | | | | | 6.433 | 3.189 | 0.569 | 0.109 | 7 | 0.654 | 5 |
| | | | | | 6.467 | 2.390 | 0.339 | 0.110 | 4 | 0.452 | 4 |
| CKM | 246 | 674 | 1551 | 3 | 2.233 | 3.481 | 0.212 | 0.010 | 6 | 0.722 | 8 |
| | | | | | 2.446 | 4.504 | 0.211 | 0.011 | 14 | 0.740 | 8 |
| | | | | | 2.219 | 3.669 | 0.241 | 0.010 | 10 | 0.759 | 8 |
| Lazega | 71 | 211 | 2571 | 3 | 12.563 | 2.243 | 0.365 | 0.179 | 6 | 0.281 | 3 |
| | | | | | 8.333 | 2.505 | 0.347 | 0.123 | 7 | 0.369 | 4 |
| | | | | | 15.549 | 1.886 | 0.341 | 0.222 | 4 | 0.301 | 3 |
| Elegans | 279 | 791 | 5860 | 3 | 4.063 | 4.523 | 0.128 | 0.016 | 12 | 0.638 | 11 |
| | | | | | 6.304 | 3.436 | 0.115 | 0.024 | 9 | 0.483 | 9 |
| | | | | | 11.486 | 2.749 | 0.207 | 0.041 | 7 | 0,432 | 6 |
| DkPol | 490 | 839 | 20198 | 3 | 4.321 | 3.965 | 0.176 | 0.020 | 9 | 0.663 | 8 |
| | | | | | 2.321 | 4.404 | 0.011 | 0.017 | 11 | 0.636 | 16 |
| | | | | | 79.922 | 1.920 | 0.520 | 0.163 | 4 | 0.183 | 6 |
| ArXiv | 14489 | 26796 | 59026 | 13 | 3.899 ± 0.945 | 5.598 ± 2.618 | 0.650 ± 0.180 | 0.03 ± 0.002 | 14.538 ± 7.523 | 0.942 ± 0.04 | 295.693 ± 154.621 |

**Table 6: Main structural characteristics of the synthetic networks; deg, apl and cc correspond to average degree, average path length, and clustering coefficient, resp.**

| $\|\mathcal{V}\|$ | $\|V_{\mathcal{L}}\|$ | $\|E_{\mathcal{L}}\|$ | deg | $\beta = 0.1$ | | $\beta = 0.5$ | |
|---|---|---|---|---|---|---|---|
| | | | | apl | cc | apl | cc |
| 500 | 1500 | 1400 | 6.667 | 12.895 | 0.321 | 9.981 | 0.061 |
| 1000 | 3000 | 14000 | 9.333 | 5.817 | 0.457 | 3.983 | 0.080 |
| 1500 | 4500 | 25500 | 11.333 | 4.929 | 0.481 | 3.629 | 0.086 |
| 2000 | 6000 | 44000 | 14.667 | 4.245 | 0.503 | 3.298 | 0.090 |
| 2500 | 7500 | 62500 | 16.667 | 4.073 | 0.507 | 3.228 | 0.088 |
| 3000 | 9000 | 87000 | 19.333 | 3.902 | 0.514 | 3.134 | 0.090 |
| 3500 | 10500 | 112000 | 21.333 | 3.819 | 0.519 | 3.091 | 0.092 |
| 4000 | 12000 | 148000 | 24.667 | 3.650 | 0.523 | 2.996 | 0.093 |
| 4500 | 13500 | 180000 | 26.667 | 3.588 | 0.522 | 2.965 | 0.092 |

where each shared neighbors is logarithmically penalized by its degree. The main assumption is that node with low degree are more informative, thus they are assigned more weight.

**Resource allocation index (RA).** [48] Similarly to Adamic-Adar, RA weighs the contribution of the shared neighbors using a heavier down-weighting factor:

$$RA(v, u) = \sum_{w \in \Gamma_v \cap \Gamma_u} \frac{1}{|\Gamma_w|}. \tag{18}$$

Compared with Adamic-Adar, it penalizes nodes with high degree more.

**Preferential-attachment (PA).** [3] measures the likelihood of connection with the product of node degrees:

$$PA(u, v) = |\Gamma_v| \cdot |\Gamma_u|. \tag{19}$$

In this case, the probability of link formation between two nodes increases as the degree of the pair of nodes.

**SimRank.** [20] Differently from the previous approaches which are local methods, SimRank is a global similarity index (i.e., it uses the whole network information), assuming that two nodes are similar if they are connected to similar nodes [24]. It is recursively defined as:

$$sr(u, v) = \omega \frac{\sum_{w \in \Gamma_v} \sum_{q \in \Gamma_u} sr(w, q)}{|\Gamma_v| \cdot |\Gamma_u|} \tag{20}$$

where $sr(u, v) = 1$ if $u = v$ then, and $\omega$ is a damping factor between 0 and 1.

## E.2 Implementation details

We implemented our method using PyTorch[3] and DGL[4] libraries. For the implementation of heuristic algorithms and MAA measure, we used the NetworkX library[5]. For SEAL [6], Neo-GNN[7], CrossMNA[8], ML-GAT[9] and GATNE[10], we used their publicly available source code. Regarding the remaining methods (MELL, Pujari, Jalili, Hristova, MAGMA) we adopted the implementation provided by [14].[11]

## E.3 Hyper-parameters

**Single layer methods.** We trained learnable single-layer approaches, i.e., Neo-GNN, and SEAL, by adopting the default hyper-parameters provided in their source code, except for the number of epochs, which was set to 100 in order to be consistent with our training procedure.

---

[3]https://pytorch.org/
[4]https://www.dgl.ai/
[5]https://networkx.org/documentation/stable/index.html
[6]https://github.com/facebookresearch/SEAL_OGB
[7]https://github.com/seongjunyun/Neo-GNNs
[8]https://github.com/ChuXiaokai/CrossMNA
[9]https://github.com/lorenzozangari/ML_GNN
[10]https://github.com/THUDM/GATNE
[11]https://www.michelecoscia.com/?page_id=1857

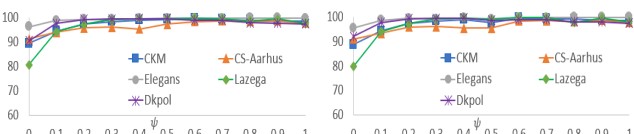

**Figure 5: AUC (left) and AP (right) values by varying $\psi$.**

**Multilayer methods.** Regarding MAGMA, for the common datasets between our work and theirs (i.e., CKM, Cs-Aarhus, Elegans), we used the same hyper-parameters provided in their paper (i.e., maximum pattern size equals to 4, confidence was 0, and support equals to 15, 20, 75, resp.). For ArXiv, we set the maximum pattern size, confidence and support equal to 4, 0 and 5. Regarding the other two datasets (Lazega and DkPol), we reduced the maximum pattern size due to out-of-time issues. For these networks, we used maximum pattern size, confidence and support equal to 3, 0 and 75, resp. For MELL, Jalili, Hristova, and Pujari, we adopted the same configuration used by [14]. For MAA score, we implemented the same formulation provided in [2], and set the relative weight of each type of triadic relation to 1. Concerning GATNE and CrossMNA, we used the default hyper-parameters provided in their source code, with the exception of the number of epochs, which was set to 100 in order to be consistent with our training procedure. In the case of CrossMNA, where the default batch size was larger than the size of some networks, we trained the model using full-batch size.

Concerning our ML-Link, for the ISL module we used the same hyper-parameters as Neo-GNN. Then, we chose $P(l) = L \setminus l$, $\psi = 0.5$, $\eta^{(\tau)} = 1$ for each $\tau$ and 1 hidden layer for all the MLPs we used. For the other hyper-parameters, we performed hyper-parameter tuning with grid search (cf. source code associated with this submission). For ML-GAT, we selected the same hyper-parameters as the GNN-NE module.

**Efficiency analysis.** Regarding the hyper-parameters chosen in the efficiency analysis, for MAGMA we selected confidence, support and maximum pattern size equal to 0, 15 and 3, resp. Although a support equal to 4 would have allowed to achieve better performance [14], we experimented out-of-memory issues when we tested MAGMA with a support value of 4.

Concerning our ML-Link, we used the ISL w/ GNN-NE version, for which we adopted the same hyper-parameters used in the comparative evaluation, except for the GNN-NE module and the number of epochs. For the former, we used hidden dimension equal to 256, attention dropout 0.7 and 1 head of attention. For the latter, we selected a value allowing to achieve comparable performance with MAGMA. Regarding the set of networks generated with $\beta = 0.1$ we used 10 epochs, while for the other set ($\beta = 0.5$) where networks are less regular, we need a higher number of epochs (20) to converge.

### E.4 Environment

We conducted all the experiments on a Linux machine (OS Ubuntu 22.04 LTS), equipped with 256GB of memory, processor Intel(R) Xeon(R) Gold 6258R CPU, 2.70GHz and GPU NVIDIA GeForce RTX 3090 with 24GB memory.

**Table 7: AUC (on the top) and AP (on the bottom) values achieved by baseline methods on real world networks. Best values are in bold.**

| Baseline | Cs-Aarhus | CKM | Elegans | Lazega | DkPol | ArXiv |
|---|---|---|---|---|---|---|
| CN | 88.104 | 73.276 | 78.509 | 79.286 | 89.743 | 98.641 |
|  | 85.375 | 71.692 | 76.58 | 77.005 | 90.309 | 98.584 |
| PA | 65.255 | 63.92 | 68.435 | 69.135 | 87.783 | 59.685 |
|  | 67.876 | 60.946 | 68.782 | 67.696 | 89.274 | 65.91 |
| Jaccard | 87.579 | 73.11 | 77.144 | 81.562 | 87.897 | 98.665 |
|  | 86.464 | 71.565 | 73.332 | 79.627 | 88.154 | 98.661 |
| RA | **89.831** | 73.511 | 79.939 | **81.860** | **92.124** | 98.698 |
|  | **89.520** | 72.831 | **79.759** | **80.398** | **92.423** | 98.702 |
| Adamic-Adar | 89.667 | **73.528** | 79.597 | 80.441 | 90.339 | 98.7 |
|  | 89.309 | **72.906** | 79.539 | 79.181 | 90.909 | 98.696 |
| SimRank | 84.829 | 70.584 | **80.322** | 61.673 | 57.803 | **99.171** |
|  | 81.524 | 70.139 | 75.558 | 56.549 | 51.560 | **99.293** |

**Table 8: AUC (on the top) and AP (on the bottom) of our method when using dot product and cosine similarity as similarity function for producing link existence scores. Best values are in bold.**

| Method | Cs-Aarhus | CKM | Elegans | Lazega | DkPol |
|---|---|---|---|---|---|
| NN-NPN (w/ *dot*) | 93.213 | 96.100 | 97.367 | 95.422 | 93.528 |
|  | 92.806 | 93.653 | 97.532 | 95.615 | 93.688 |
| NN-NPN (w/ *cos*) | **95.927** | **98.561** | **98.828** | **99.023** | **98.036** |
|  | **95.481** | **97.576** | **98.865** | **99.104** | **98.560** |

## F ADDITIONAL RESULTS

### F.1 Baseline results

Table 7 shows the AUC and AP values achieved by each baseline method. RA shows the best average results across all datasets, while PA is the worst performing approach.

### F.2 Sensitivity analysis with the full framework

Figure 5 shows the AUC and AP values achieved by the whole framework (i.e., using both the NN-NPN and the GNN-NE component). We can observe the same trend as for the NN-NPN component, but with better performance, particularly for the minimum and maximum values of $\psi$, i.e., 0 and 1. Thus, we can conclude that ML-Link in its entirety is more robust w.r.t. the choice of $\psi$ than using only the NN-NPN component.

### F.3 Node-pair similarity analysis

We also present an empirical comparison between the NN-NPN component using cosine similarity and dot product, which is employed by [42], as similarity functions for producing the similarity scores of links existence (Eqs. 8 and 9 in the main text). Table 8 shows that, when using cosine similarity, ML-Link achieves better AUC and AP scores in all cases. This could be due to the normalization factor, which helps in controlling the magnitude of vector representations, thus making the training process more stable.

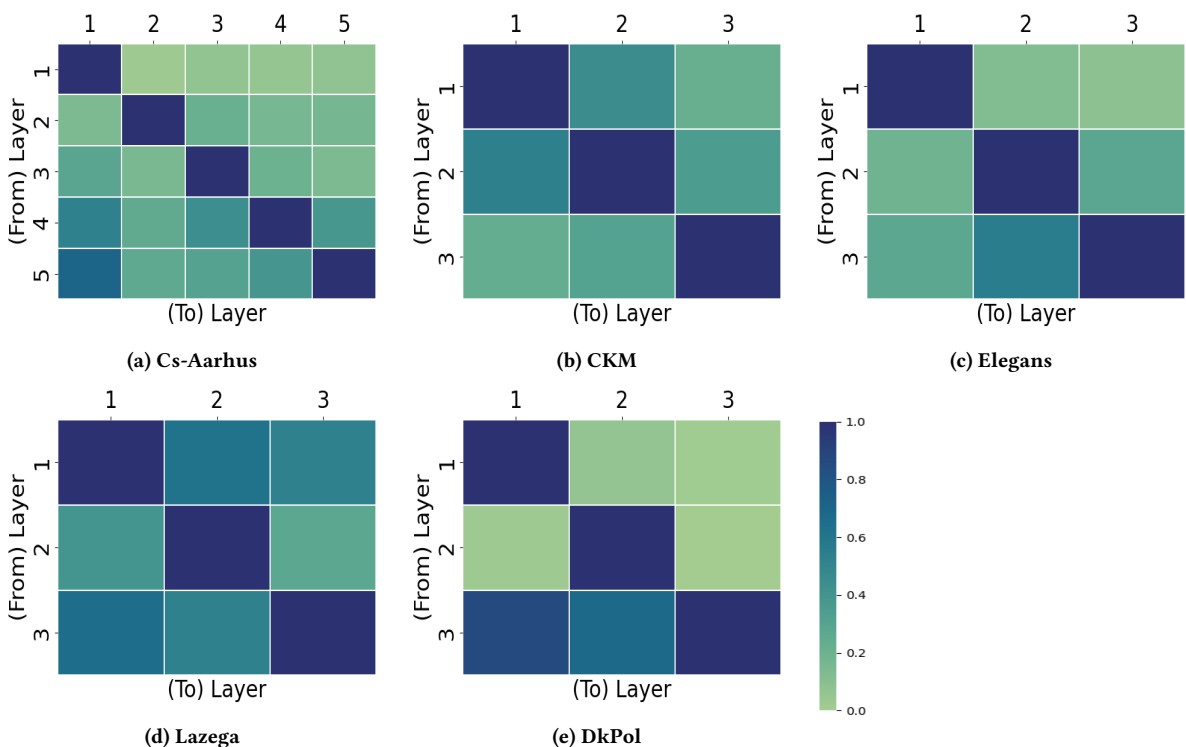

**Figure 6: Conditional probability overlap (*cpo*) for Cs-Aarhus, CKM, Elegans, Lazega, DkPol (left to right). Darker colors correspond to higher *cpo*.**

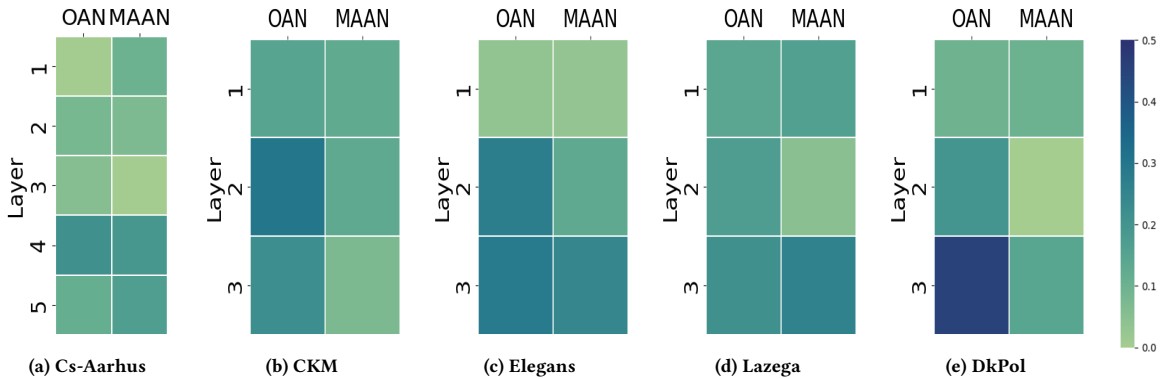

**Figure 7: Attention coefficients learned for Cs-Aarhus, CKM, Elegans, Lazega, DkPol (from left to right). Darker colors correspond to higher attention intensity.**

## F.4 Visual interpretation of the attention coefficients

We visualize the attention coefficients extracted by the CLA module (cf. Eq. 11 in the main text) which, for each layer, learns the importance of the predictive information yielded by each overlapping multilayer neighborhoods, i.e., $\Gamma^{(oan)}$ and $\Gamma^{(maan)}$.

To support our analysis, we compute the similarity between layers employing the conditional probability overlap (*cpo*), defined as follows [4]:

$$\omega(l, l') = \frac{nonzero(\mathbf{A}_l \circ \mathbf{A}_{l'})}{nonzero(\mathbf{A}_{l'})} \tag{21}$$

where $\omega(l, l')$ is the probability of finding a link in layer $l$, given the existing edges in layer $l'$; *nonzero* counts the number of nonzero values in a matrix, and $\circ$ is the element-wise product. Figure 6

shows the *cpo* value for each real-world network, which yields a non-symmetric similarity matrix for each dataset. We consider the entry $[l, l']$ of each matrix as the relative importance the layer $l'$ has for predicting links formation in layer $l$. However, since we used the summation operator as $\bigoplus$ (cf. Eq. 9 in the main text) — thus obtaining the overall across-layer predictive information under each $\tau$ — the attention coefficients and the *cpo* are not directly comparable. Furthermore, the former are normalized such that $\sum_{\tau \in \mathcal{T}} \sum_{l \in L} \alpha^{(\tau,l)} = 1$, while each $\omega(l, l')$ is a conditional probability value for each pair of layers.

Consequently, to interpret the attention weights we use the *cpo* score as a benchmark. That is, we expect that the *cpo* score and the magnitude of the attention coefficients follow similar patterns. Figure 7 shows a heatmap expressing the attention coefficients for each dataset. This shows how each layer distributes its attention over different $\tau \in \mathcal{T}$. As discussed, the attention mechanism allows the model to selectively integrate the overall predictive information produced by each overlapping multilayer neighborhoods for each layer. For example, in Lazega, the model assigns almost the same importance to each $\tau$ in layer 1 (Advice), but it assigns a larger magnitude to the OAN and the MAAN contexts in layers 2 (Friendship) and 3 (Co-Workship), respectively. Similarly in CKM, the model assigns similar weights to each context in layer 1 (Advice), but assigns greater weights to the OAN context in the remaining two layers (Discussion and Friendship). For Cs-Aarhus, the model prefers to alternate the importance that is given to the different contexts at each layer. Interestingly, the model prefers the MAAN context for Co-authorship (1) and Work (5) layers, which can extract more meaningful and deeper connections. However, the model prefers the OAN context for Leisure (3) and Lunch (4) layers . For DkPol, the OAN context is weighted heavily, especially in layer 3 (Follow).

ML-Link can distribute the attention not only across different multilayer neighborhoods, but also across the layers, thus effectively estimating the importance of across-layer predictive information.

Comparing Figs. 6 and 7, we can observe that the attention weights have higher magnitudes for layers with higher *cpo*, and lower magnitudes for layers with lower *cpo*. This is as expected, since the *cpo* score measures the similarity between layers, and we would expect layers that are more similar to be more important for each other. For example, in DkPol layers 1 and 2 are relatively important for layer 3, as shown in Fig. 6. Similarly, in Fig. 7 we can observe that for layer 3 the predictive information carried by the ESL module is weighted with high intensity. Considering Elegans and Cs-Aarhus networks we observe a similar behavior, where layers with higher similarity according the *cpo* score are weighted heavily by the attention mechanism. Conversely, in CKM and Lazega, all layers exhibit a high demand for external information from other layers (Fig. 6). In these cases, the attention weights are distributed more uniformly (Fig. 7) across all the layers. This shows that ML-Link is able to adapt its attention mechanism to the specific characteristics of the network.