# OpenReview forum: "Link Prediction on Multilayer Networks through Learning of Within-Layer and Across-Layer Node-Pair Structural Features and Node Embedding Similarity"
_ACM.org/TheWebConf/2024/Conference — TheWebConf24_

### Official Review · Reviewer_ggFv · 2023-11-09

**Novelty:** 5
**Technical Quality:** 5

**Review:**

The authors propose a neural-network-based approach for link prediction in multilayer networks which they call ML-Link.
They combine GNN-based node embeddings, intra-layer node similarity, and oberlapping inter-layer node neighbourhoods using an attention mechanism.
Applied to a set of real networks, the authors find that their approach outperforms current baseline methods, achieving high AUC and AP values around 99% on average.
Upon examining the learned attention weights, the authors find that their proposed method assigns different attention weights to different layers.
A hyperparameter controls how much weight should be put on layer-internal vs. -external features, however, the authors show that their method is not very sensitive to the setting of this parameter, achieving similar results over a relatively large range for values of this hyperparameter.

The paper is well-written and follows a simple idea: to use observed link patterns across multiple layers to inform link prediction.
The overall approach is well-motivated and all mathematical symbols used in the text are defined.
However, at times it seems like the authors could express the same ideas in a slighly less formal way to help the readers follow their story.
I believe this paper will be useful for the research community and practitioners.
My main concern about the paper is the relatively small number of quite small networks that were used for evaluation, and I have several questions, detailed below.

**Questions:**

1. ML-Link was evaluated mainly on quite small networks with a small number of layers, except for the arXiv network with close to 15,000 nodes. What is the reason for this?
2. Could ML-Link be applied for link prediction in single-layer networks? I suppose this would be the same as saying that one of the layers should not be paired with any other layer. How does ML-Link perform in this case?
3. Did I understand it right that ML-Link operates on unweighted and undirected networks? If so, how is ML-Link used to predict links for the directed datasets?
4. The results show that ML-Link performs very well on networks from different domains. The authors mention that some of the used baseline methods do not perform well on the Lazega network because of its structural properties. Are there any such limitations that apply to ML-Link? That is, what structural properties make it harder for ML-Link to perform well?
5. ML-Link uses node metadata, if available, to compute node embeddings with GNNs. Could metadata labels also be used to further inform internal and external structure learning?

**Ethics Review Description:**

-

**Reviewer Confidence:**

3: The reviewer is confident but not certain that the evaluation is correct

**Scope:**

4: The work is relevant to the Web and to the track, and is of broad interest to the community

---

### Official Review · Reviewer_UoJr · 2023-11-20

**Novelty:** 4
**Technical Quality:** 5

**Review:**

The paper aims to fill a lack of multilayer graph representation learning methods designed for link prediction. The model considers the link prediction task very well. It not only considers the similarity between nodes, but also considers the influence of graph structure on link prediction tasks from the layer-level perspective. The performance of the model on multilayer graph link prediction and the effectiveness of cross-layer are demonstrated through extensive experimental results.

## Strength
1. The paper is clearly written and describes the problem definition in detail.
2. The paper is well-considered for link prediction and the experimental results are very good in performance.
3. The paper provides a complete implementation setting and evaluation baseline.

## Weakness
1. Novelty. The Across-Layer feature propagation has been proposed in GNN with many solutions, and the solution in this paper is trivial, I hope the authors can compare the related methods in more depth, and the challenges in it.
2. Redundant feature propagation. There are inevitably many redundant information propagations between ML-GAT and ISL, ESL in the model. When the propagation meets noise nodes, there will be a large extent through the cross-layer diffusion. Even if there are no early noise effects, it also has many unnecessary redundant propagations. Hope to be further optimized.
3. Complexity. The model not only needs to train a GNN model to deal with pairwise node similarity, but also needs to add an extra NN module for information propagation from layer level. Learning across layers requires more detailed experiments on the order parameters. we are unable to judge the superiority of the improved method compared to other end-to-end methods in terms of time and memory, especially when the graph size is relatively large.

**Questions:**

See Review.

**Reviewer Confidence:**

3: The reviewer is confident but not certain that the evaluation is correct

**Scope:**

4: The work is relevant to the Web and to the track, and is of broad interest to the community

---

### Official Review · Reviewer_J3hv · 2023-11-24

**Novelty:** 6
**Technical Quality:** 6

**Review:**

The paper presents a neural network-based learning framework for link prediction on (attributed) multilayer networks. The authors claim that it is the first work to augment multilayer GNNs with node-pair features learned from both within-layer and across-layer
structural information. The experiments demonstrate the effectiveness of the proposal.

**Questions:**

The major concern is the baselines. Whether the baselines are strong competitors is not clearly introduced. In addition, the proposal can outperform the other baselines by a large margin in CKM, but the result is not well explained.

**Reviewer Confidence:**

3: The reviewer is confident but not certain that the evaluation is correct

**Scope:**

4: The work is relevant to the Web and to the track, and is of broad interest to the community

---

### Official Review · Reviewer_o3yt · 2023-11-27

**Novelty:** 3
**Technical Quality:** 3

**Review:**

Summary:
This paper discusses the problem of link prediction in multilayer networks and proposes a novel neural-network-based learning framework called ML-Link. The framework combines pairwise similarities of multilayer node embeddings learned by a graph neural network model with structural features learned from within-layer and across-layer link information based on overlapping multilayer neighborhoods.

Pros：
1. This paper is well-written and easy to follow.

2. The method proposed in the paper is novel. ML-Link is the first to propose augmenting multilayer GNNs with node-pair features learned from both within-layer and across-layer structural information. The authors consider comprehensively the internal structure of a target layer, the structure of the other layers in a network, and the layer-specific node-attributes.

3. Experimental verification is comprehensive, involving a diverse range of datasets, performance metrics, and evaluation methodologies.


Cons:
1. Baselines are not new enough and should be considered with some recent studies on multiplex networks.

2. The model diagram is a bit unclear and intuitive.

**Questions:**

1. Since the connection existence of each module is calculated through the cosine similarity between representations, can some multilayer network representation learning methods be considered in baselines?  Some references are as follows：DMGI[1], BPHGNN[2], DualHGNN[3], HDMI[4] and MHGCN[5].
[1] Park, Chanyoung, et al. "Unsupervised attributed multiplex network embedding." Proceedings of the AAAI Conference on Artificial Intelligence. Vol. 34. No. 04. 2020.

[2] Fu, Chaofan, et al. "Multiplex Heterogeneous Graph Neural Network with Behavior Pattern Modeling." Proceedings of the 29th ACM SIGKDD Conference on Knowledge Discovery and Data Mining. 2023.

[3] Xue, Hansheng, et al. "Multiplex bipartite network embedding using dual hypergraph convolutional networks." Proceedings of the Web Conference 2021. 2021.

[4] Jing, Baoyu, Chanyoung Park, and Hanghang Tong. "Hdmi: High-order deep multiplex infomax." Proceedings of the Web Conference 2021. 2021.

[5] Yu, Pengyang, et al. "Multiplex heterogeneous graph convolutional network." Proceedings of the 28th ACM SIGKDD Conference on Knowledge Discovery and Data Mining. 2022.

2. In Table 1, one might question the seemingly exceptional performance of ML-Link in terms of AUC and AP across numerous datasets, i.e., most results are more than 99%.

Please see the above cons.

**Reviewer Confidence:**

4: The reviewer is certain that the evaluation is correct and very familiar with the relevant literature

**Scope:**

4: The work is relevant to the Web and to the track, and is of broad interest to the community

---

### Official Review · Reviewer_vieH · 2023-12-01

**Novelty:** 5
**Technical Quality:** 5

**Review:**

The authors propose the ML-Link approach for link prediction on multi-layer networks with node attributes. There are two main components: a multi-layer graph neural network (GNN) and node pair features learned from within- and across-layer structural information. The latter is their primary new contribution. Experiment results shown extremely impressive gains in accuracy compared to other link prediction methods, including others designed for multi-layer networks.

*After author rebuttal:* The authors clarified their reasoning for choosing data sets, and I do agree that maintaining data sets used in prior work is an important thing to make fair comparisons. To potentially improve the paper, perhaps the authors could also try to create some more challenging setting (e.g., by taking negative samples differently).

## Strengths
- Extremely strong empirical performance across multiple data sets. Unlike most papers that report a small increase over competing methods, the improvement offered by ML-Link is often so large that it seems almost unbelievable in some cases.
- The nearest-neighbor-based node-pair neighborhood (NN-NPN) feature extraction component looks to be novel, particularly the external structure learning and context-level attention components.
- Very well written and comprehensive paper with a good balance of technical details and results.

## Weaknesses
- The extremely high accuracy values achieved suggest that the experimental evaluation approach may be too easy (see question 1 below).
- Heavy use of acronyms throughout the paper that makes it difficult to keep up at times. For example, in Table 2, instead of acronym soup, perhaps it would be better to write Internal and External instead of ISL and ESL.

**Questions:**

1. In most of the data sets, your ML-Link method achieves 99+% in the area under the ROC curve (AUC) and average precision (AP) metrics. This is extremely high and far beyond what I would expect to be achievable for so many data sets. How should we interpret these results? Are the data sets perhaps too easy, or is it due to the negative sampling procedure resulting in negative samples that are too easy?
2. What are some limitations and areas for future improvement that you see? Please include these in your conclusion for the benefit of the reader.
3. Are the three loss functions $\mathcal{I}\_{npn}$, $\mathcal{I}\_{ne}$, $\mathcal{I}\_{p}$ directly summed or first weighted with tunable weights?

**Ethics Review Description:**

No concerns

**Reviewer Confidence:**

3: The reviewer is confident but not certain that the evaluation is correct

**Scope:**

3: The work is somewhat relevant to the Web and to the track, and is of narrow interest to a sub-community

---

### Decision · Program_Chairs · 2024-01-22

**Decision:**

Accept

**Comment:**

The consensus is that this is a good paper with strong experimental results. There were some discussions on whether the experimental setting was somehow too easy. (But it is consistent with previous work, so there is a clear improvement in this paper.)
 One reviewer felt that the data sets themselves are too easy, but that the random negative sampling approach is making the link prediction task too easy.

 If possible, it would be good to have some comments on this in the camera-ready version.